# The Impact of Kiwifruit Consumption on the Sleep and Recovery of Elite Athletes

**DOI:** 10.3390/nu15102274

**Published:** 2023-05-11

**Authors:** Rónán Doherty, Sharon Madigan, Alan Nevill, Giles Warrington, Jason Gordon Ellis

**Affiliations:** 1Sports Lab North West, Atlantic Technological University Donegal, Letterkenny Campus, Port Road, F92 FC93 Letterkenny, Ireland; 2Sport Ireland Institute, National Sport Campus, Abbotstown, D15 PNON Dublin, Ireland; smadigan@instituteofsport.ie; 3Northumbria Centre for Sleep Research, Northumbria University, Newcastle NE7 7XA, UK; jason.ellis@northumbria.ac.uk; 4Sport and Human Performance Research Centre, University of Limerick, V94 T9PX Limerick, Ireland; giles.warrington@ul.ie; 5Department of Physical Education and Sport Sciences, University of Limerick, V94 T9PX Limerick, Ireland; 6Faculty of Education, Health and Wellbeing, University of Wolverhampton, Walsall Campus, Walsall WV1 1LY, UK; a.m.nevill@wlv.ac.uk

**Keywords:** sleep, nutrition, recovery, athletes

## Abstract

Background: Poor sleep and resultant under-recovery can negatively impact training adaptations, increase the risk of injury and reduce subsequent performance. Due to the ‘food first’ approach adopted by many athletes, there is scope for investigation of ‘functional food’ based interventions (i.e., kiwifruit contains melatonin which plays a role in circadian rhythm regulation) designed to promote athlete recovery and/or enhance sleep quality and quantity. Methods: Following the baseline assessment (Week 1) all subjects began the intervention (Weeks 2–5). During the 4-week intervention, participants were asked to consume 2 medium-sized green kiwifruit (*Actinidia Deliciosa*) an hour before bed. Participants completed a questionnaire battery at baseline and post-intervention, and a daily sleep dairy for the duration of the study. Results: The results demonstrated a positive impact of kiwifruit consumption on key aspects of sleep and recovery in elite athletes. From baseline to post-intervention, there were clinically significant improvements in sleep quality (i.e., improved PSQI global scores and sleep quality component scores) and improvements in recovery stress balance (reduced general stress and sports stress scales). Moreover, the intervention improved sleep as evidenced by significant increases in total sleep time and sleep efficiency % and significant reductions in number of awakenings and wake after sleep onset. Conclusion: The findings broadly suggested that kiwifruit does impact positively on sleep and recovery in elite athletes.

## 1. Introduction

Elite athletes are predisposed to challenges to their sleep, such as habitual short sleep duration (<7 h per night) and poor sleep quality (e.g., fragmented sleep) [1]. The multifaceted demands placed on elite athletes including the frequency, volume, intensity and timing of training and competition [1,2,3], performance anxiety [4,5] and travel requirements [6,7] can all negatively impact sleep health. Good sleep health is characterised by satisfaction, appropriate timing, adequate duration, high efficiency, and sustained alertness during waking hours and can be assessed in athletes using the Regulatory, Satisfaction, Alertness, Timing, Efficiency and Duration questionnaire (RU-SATED) [8]. Sleep health is a concept which involves a holistic view of sleep as opposed to individual symptoms and disorders [9]. Other lifestyle factors (e.g., nutrition, caffeine use) and exposure to technology (i.e., blue light exposure at night), can also have a detrimental impact on athletes’ sleep [1]. Poor sleep and resultant under-recovery can negatively impact training adaptations, increase the risk of maladaptation and reduce subsequent performance [10], which may lead to non-functional overreaching (NFO) in the shorter term and over-training syndrome (OTS)/unexplained underperformance syndrome (UUPS) in the longer term [11,12].

The health benefits of consuming fruit are well documented [13]. It has previously been suggested that athletes’ sleep could be improved by analysing and improving their eating habits by, for example, increasing their intake of fruit, vegetables and fish while reducing their intake of processed foods [14]. Given the adoption of a ‘food first’ approach by many athletes [15], there is scope for investigating food-based interventions designed to promote athlete recovery and/or enhance sleep health. Antioxidants are commonly consumed by athletes in an attempt to reduce oxidative stress following training.

Kiwifruit have also been shown to contain melatonin (24 µg/g) [16], which plays an important role in circadian rhythm regulation, i.e., getting to sleep and maintaining sleep are easiest at and after the onset of melatonin secretion. The serotonin (5.8 µg/g) content in kiwifruit may contribute to improved sleep while the rich antioxidant content may supress free radical expression and inflammatory cytokines. Folate deficiency has been linked to insomnia and restless leg syndrome; the folate in kiwifruit may improve folate status and consequently improve sleep [17]. A small, randomised crossover study comprising 6 males and 8 females demonstrated that the consumption of varying doses of kiwifruit (1–3/d × 3 weeks, with 2-week washout between doses) resulted in a significant increase in plasma Vitamin C levels [18]. Compared to the baseline, consumption of 2 kiwifruit daily significantly raised plasma Vitamin C levels by 20% (73 µM ± 4; *p* < 0.01) [18]. Additionally, improved antioxidant status was evident; lymphocytes isolated from blood collected from participants demonstrated decreased sensitivity to oxidative attack by (H_2_O_2_) in vitro, and endogenous oxidation of lymphocyte DNA was also decreased [18]. However, it must be noted that the results of this study may not be generalizable due to the very small sample size.

In the last decade, kiwifruit has received attention in terms of potential sleep promoting properties. To date, the research has focused on populations either self-reporting or having a diagnosed sleep problem. A study involving volunteers (*n* = 25) who self-reported sleep disturbance demonstrated that consumption of 2 kiwifruit 1 h before bedtime for 4 weeks significantly improved actigraphy-measured total sleep time (16.9%) and sleep efficiency (2.4%) (*p* < 0.001) [17]. Self-report measures of sleep also improved significantly; WASO reduced (−28.9%); and sleep onset latency reduced (−35.4%) while sleep efficiency increased (5.4%) (*p* ≤ 0.002) [17]. Sleep quality and duration were significantly improved following the 4-week kiwifruit intervention. However, it must be noted that sleep was not monitored during the intervention period. In a similar study, students (*n* = 74) with diagnosed insomnia (using the Bergen Insomnia scale) consumed either 130 g of kiwifruit or a placebo (130 g pear) 1 h before bed for 4 weeks, and sleep was assessed by both actigraphy and sleep diaries. While there were no statistically significant differences in objective measures of sleep, there were statistically significant group x time effects for subjective sleep quality (F1, 51 = 5.88, *p* < 0.05) and daytime function (F1, 51 = 4.79, *p* < 0.05) [19]. These promising findings warrant further investigation within athletic populations in relation to kiwifruit consumption and their interaction with sleep and recovery. Further research is necessary to investigate the potential benefits and practical application of kiwifruit consumption to promote post-exercise recovery and/or improve sleep quality and quantity in athletes.

The current study is the first step in the development of a specific ‘whole-food’ nutritional intervention for optimising sleep quality, sleep quantity, sleep health and/or enhancing post-exercise recovery in elite athletes. This is the first study to investigate the impact of kiwifruit consumption on the sleep and recovery of elite athletes. The aims of this study are the following:To characterise the baseline sleep and recovery levels of elite athletes;To assess the impact of kiwifruit supplementation on the sleep and sleep health of elite athletes;To reassess the sleep and recovery levels of elite athletes after the 4-week intervention.

## 2. Materials and Methods

### 2.1. Design

This study was an open label trial of the impact of kiwifruit supplementation on sleep and recovery in athletes, and as such, the purpose of the trail was not withheld from participants [20]. The participants were told that the purpose of the research was to investigate the impact of kiwi ingestion on athlete recovery.

### 2.2. Participants

A group of elite athletes (*n* = 15) from a national sailing squad (*n* = 9; 7 males and 2 females) and a national athletics squad (middle distance runners; *n* = 6; 2 males and 4 females; see Table 1) were recruited through the National Sports Institute. Athletes were regarded as elite in line with published definitions, i.e., members of a national team [21]. No participants reported using sleep medication at baseline or post-intervention.

### 2.3. Procedure

All procedures were approved by the research ethics committee of the Faculty of Health and Life Sciences, Northumbria University. After reading the participant information sheet, all participants provided written informed consent prior to data collection. The participants were the provided with a link to the baseline and post-intervention questionnaire battery and the daily sleep diary. Participants were instructed to complete the baseline questionnaire battery, commence completion of the daily sleep diary for the duration of the study (5 weeks; 1 control week and 4 intervention weeks) and complete the post-intervention questionnaire battery upon completion of the study, i.e., after week 5 (see Figure 1).

### 2.4. Measures

All participants completed demographic data before completing the baseline and post-intervention questionnaires. Participants recorded their sex, age, body mass (kg), height (cm), sport, athlete type (elite or sub-elite), phase of season (pre-season, competition or off-season), normal training time (before 8 a.m., 8 a.m. to 5 p.m. and after 5 p.m.) and training/competition duration per week (mins) (see Table 1).

### 2.5. The Recovery Stress Questionnaire for Athletes (RESTQ Sport)

High scores on the stress scales indicate a high level of stress, while high scores on the recovery scales indicate a high level of recovery [22]. Each item is scored on a Likert scale (from 0 = Never to 6 = Always) based on how often the respondent engaged in a specified activity over the previous three days/nights, with a response of 0 indicating never having experienced the feeling and 6 indicating always experiencing the associated feeling. High scores on stress scales indicate a high level of stress, while high scores on the recovery scales indicate a high level of recovery [22].

### 2.6. Pittsburgh Sleep Quality Index (PSQI)

The PSQI is a self-report measure of sleep quality, consisting of 19 items grouped into 7 component scores which are equally weighted [23]. Overall global scores (GPSQI) were calculated by summing the seven components (range 0–21, with higher scores indicating poorer sleep quality), and the component scores were also calculated to provide subscale ratings of (i.) subjective sleep quality, (ii.) sleep latency, (iii.) sleep duration, (iv.) sleep efficiency, (v.) sleep disturbances, (vi.) use of sleep medication and (vii.) daytime dysfunction [24]. As athletes often strive for marginal gains in their performance, which can be facilitated through optimised sleep, the identification of both ‘poor’ and ‘moderate’ sleep quality is warranted [25]; hence, the standard cut-off (≥5) was employed for GPSQI.

### 2.7. Consensus Sleep Dairy-Core (CSD-C)

The CSD-C, a standardised sleep diary, and the data collected were used to compute indices of sleep continuity such as sleep onset latency (SOL), number of awakenings (NoA), wake after sleep onset (WASO), time in bed (TIB), total sleep time (TST) and sleep efficiency (SE) [26]. Additional Likert scales were used to report fatigue both before going to bed and on getting up in the morning (from 1 = Completely Exhausted to 8 = Fully Alert) and sleep quality (from 1 = Very Poor to 5 = Very Good). There was also a question relating to adherence where the participants recorded if they consumed kiwifruit or not each day.

### 2.8. The Regulatory, Satisfaction, Alertness, Timing, Efficiency and Duration (RU-SATED)

#### 2.8.1. Questionnaire

The RU-SATED was developed to assess Sleep Health [8]. Sleep health is identified through regulation, satisfaction, appropriate timing, adequate duration, high efficiency and sustained alertness during waking hours [8]. Sleep health is a concept which involves a holistic view of sleep as opposed to individual symptoms and disorders [9]. The RU-SATED assesses six dimensions of sleep health:Regulation: consistent sleep–wake schedule (within 1 h)Satisfaction/quality: subjective assessment of ‘good’ or ‘poor’ sleepAlertness/sleepiness: ability to maintain wakingTiming: placement of sleep within the 24 hEfficiency: ease of falling asleep and returning to sleepDuration: total amount of sleep per 24 h [8].

Each dimension is scored on a 3-point Likert scale from 0 (rarely/never) to 2 (usually/always); the scores from each dimension can be converted to a total score (0–12) with higher scores indicative of good sleep health [27].

#### 2.8.2. Kiwifruit Intervention

Following the baseline assessment (Week 1) all subjects began the intervention (Weeks 2–5). During the 4-week intervention, participants were asked to consume 2 medium-sized green kiwifruit (*Actinidia Deliciosa*) an hour before bed (See Figure 1). The dose was based on doses employed in previous studies (2 × Kiwi [17]) and (130 g [19]), and the timing was proposed to coincide with melatonin secretion. The participants reported adherence to the intervention when completing the daily questionnaire. As the research was conducted under ‘lockdown’ conditions during the COVID-19 pandemic, participants were instructed to purchase the kiwifruit themselves and were reimbursed upon completion of the study.

### 2.9. Data Analysis

All data were analysed using the Statistical Package for the Social Sciences (SPSS Version 26, IBM Corporation) and Jamovi (Version 1.6, The Jamovi Project). Frequency distribution and descriptive statistics were used to present findings [28]. All data were presented in mean ± standard deviation and/or frequency. Shapiro–Wilk tests were used to assess the distribution of data. Paired samples *t*-test and Wilcoxon signed rank tests were used to examine the changes in scores from baseline to post-intervention. Effect sizes were calculated using Cohen’s d and interpreted as small d ≥ 0.2, medium d ≥ 0.5 and large d ≥ 0.8. Repeated measures ANOVA and Friedman’s test were used to assess the difference in scores from baseline and week by week during the intervention. For variables that demonstrated significant differences, pairwise comparisons were performed to identify each timepoint where significant differences occurred compared to baseline.

## 3. Results

In total, 15 elite athletes took part from a national sailing squad (*n* = 9; male *n* = 7 and female *n* = 2) and a national athletics squad (*n* = 6; male *n* = 2 and female *n* = 4). An independent samples *t*-test highlighted significant differences between the groups at baseline for body mass (t = −4.931; *p* < 0.001), height (t = −2.338; *p* < 0.05) and training/competition duration per week (t = −3.066; *p* < 0.01), which is indicative of the different characteristics of sailing and athletics. However, a Wilcoxon signed rank test revealed no statistically significant differences from baseline to post-intervention for body mass, normal training time, training duration and phase of season (*p* > 0.05) (see Table 1).

### 3.1. Baseline vs. Post-Intervention (PSQI and RESTQ)

In order to investigate changes in sleep quality and recovery/stress balance over the duration of the study, participants completed the PSQI and RESTQ at baseline and post-intervention. There were no significant gender or sport effects for any measures in the current study.

#### 3.1.1. Sleep Quality

A Wilcoxon signed rank test was used to compare the PSQI component scores from baseline to post-intervention (see Table 2). Sleep quality improved significantly from baseline (1.53 ± 0.84) to post-intervention (0.27 ± 0.46; z = 78, *p* = 0.002) (see Table 2 and Figure 2). PSQI global scores reduced significantly from baseline (6.47 ± 2.17) to post-intervention (4.13 ± 1.19; z = 91, *p* = 0.002) (see Table 2 and Figure 3). This was also clinically relevant, indicating a significant reduction in sleep problems among the athletes. While there were improvements from baseline to post-intervention in sleep onset latency, sleep duration and sleep efficiency, no significant differences were observed (*p* > 0.05). Similarly, daytime dysfunction component scores did not change from baseline and post-intervention.

#### 3.1.2. Recovery

Wilcoxon signed rank tests were used to compare the baseline and post-intervention RESTQ scores. There were statistically significant improvements from baseline to post-intervention for the RESTQ scales general stress (3 ± 0.86 vs. 2.58 ± 0.58; z = 2.77, *p* = 0.015) and sport stress (2.72 ± 0.65 vs. 2.39 ± 0.63; z = 2.85, *p* = 0.019). Conversely, while both increased, there were no statistically significant differences between general recovery (3.83 ± 0.79 vs. 4.11 ± 0.84; z = −2.09, *p* = 0.71) and sport recovery (4.03 ± 1.06 vs. 4.09 ± 0.92; z = −0.4, *p* = 0.65).

A Wilcoxon signed rank test was also used to compare the 19 RESTQ sub scale items from baseline to post-intervention (see Table 3). There were statistically significant reductions in fatigue (3.4 ± 1.42 vs. 2.8 ± 1.21; z = 50.5, *p* = 0.02), physical complaints (2.37 ± 0.92 vs. 1.8 ± 0.6; z = 55, *p* = 0.005) and disturbed breaks (2.38 ± 0.8 vs. 2.03 ± 0.75; z = 57, *p* = 0.036) (see Figure 4, Figure 5 and Figure 6).

### 3.2. Intervention (Weeks 2–5)

Adherence (90 ± 6.64%; range 82.14–100%) to the kiwifruit intervention was high for all participants. The majority of athletes in this study (*n* = 10) were in pre-season which may have improved adherence. In order to investigate changes in sleep quality and sleep health, participants completed the CSD-C and RU-SATED daily for the duration of the study.

#### 3.2.1. Sleep Diary

The daily sleep diary data were averaged and analysed on a week by week basis (see Table 4). A repeated measures ANOVA and Friedmans test were used to assess the difference between the baseline and intervention weeks for the sleep diary data. The normally distributed variables (SOL, TIB and TST) were analysed using a repeated measures ANOVA, while Friedman’s test was used to analyse the non-normally distributed variables (Awakenings, WASO, SE, Fatigue and SQ). Where there were significant differences, pairwise comparisons were performed to assess if there were significant differences between variables at each timepoint during the intervention compared to baseline.

A repeated measures ANOVA demonstrated that although SOL reduced during the intervention compared to baseline, there were no statistically significant differences (F(1.81, 19.92) = 2.689, *p* > 0.05). Conversely, TIB increased from baseline to intervention, but the differences were not statistically significant (F(1, 11) = 0.393, *p* > 0.05). TST improved week to week from baseline to intervention (F(4, 44) = 6.653, *p* = 0.001 partial η^2^ = 0.38). TST increased from baseline 7.6 ± 0.75 h to 8.55 ± 0.44 h at week 4, a statistically significant increase of 0.83 ± 0.23 ([mean ± standard error], *p* < 0.05) (see Figure 7).

A Friedman’s test highlighted that NoA reduced significantly from baseline to intervention: χ^2^(4) = 12.6, *p* < 0.05. Pairwise comparisons (Durbin–Conover) were performed to assess if there were significant differences between timepoints during the intervention compared to baseline. Pairwise comparisons demonstrated that there was a statistically significant reduction in NoA compared to baseline in weeks 3 (*p* = 0.003) and 4 (*p* = 0.012) (see Figure 8).

A Friedman’s test highlighted that WASO reduced significantly from baseline to intervention: χ^2^(4) = 12.5, *p* < 0.05. Pairwise comparisons (Durbin–Conover) demonstrated that there was a statistically significant reduction in WASO compared to baseline in week 3 (*p* = 0.002), week 4 (*p* = 0.003) and week 5 (*p* = 0.014) (see Figure 9).

A Friedman’s test showed that SE increased significantly from baseline to intervention: χ^2^(4) = 21.2, *p* ≤ 0.001. Pairwise comparisons (Durbin–Conover) demonstrated that there was a statistically significant increase in SE compared to baseline in week 2 (*p* = 0.018), week 3 (*p* < 0.001), week 4 (*p* < 0.001) and week 5 (*p* < 0.001) (see Figure 10). Self-report Fatigue Going to Bed did not differ significantly from baseline to intervention, χ^2^(4) = 3.05, *p* = 0.55, while there was a significant difference in Fatigue in the Morning from baseline to intervention: χ^2^(4) = 15.6, *p* = 0.004. Pairwise comparisons (Durbin–Conover) demonstrated that there was a statistically significant reduction in Fatigue in the Morning compared to baseline in week 5 (*p* = 0.041) (see Figure 11).

While all Sleep Quality scores were higher during the intervention compared to baseline, the differences were not significant: χ^2^(4) = 8.62, *p* = 0.071.

#### 3.2.2. Sleep Health

A repeated measures ANOVA demonstrated that, although the Sleep Health scores increased during the intervention (weeks 3–5) compared to baseline, there were no statistically significant differences (F(4, 44) = 1.178, *p* > 0.05). A Friedman’s test was used to assess the difference between the baseline and intervention weeks for the RU-SATED data (see Table 5).

A Friedman’s test showed that Efficiency increased significantly from baseline to intervention: χ^2^(4) = 10.2, *p* ≤ 0.036. Pairwise comparisons (Durbin–Conover) demonstrated that there was a statistically significant improvement in Efficiency compared to baseline in week 3 (*p* = 0.005), week 4 (*p* = 0.021) and week 5 (*p* = 0.009) (see Figure 12). There were no significant differences between Regulation, χ^2^(4) = 8.62, *p* = 0.417, and Satisfaction/Quality, χ^2^(4) = 2.55, *p* = 0.637. While Alertness/Sleepiness scores improved week to week, the changes were not significant: χ^2^(4) = 8.48, *p* = 0.075. There were no significant differences between Timing, χ^2^(4) = 1.00, *p* = 0.91, and Duration, χ^2^(4) = 4.11, *p* = 0.392.

## 4. Discussion

The aims of the current study are to (i) assess the baseline sleep and recovery levels of elite athletes, (ii) assess the impact of kiwifruit supplementation on the sleep of elite athletes and (iii) reassess the sleep and recovery levels of elite athletes after the intervention. This is the first study to assess the impact of kiwifruit consumption on the sleep and recovery of elite athletes. To date, very limited research exists investigating the potential sleep promoting properties of kiwifruit, and this study is the first to investigate the impact of kiwifruit consumption on the sleep and recovery of elite athletes. As such, further research is necessary to develop practical guidelines for supplementation to enhance sleep and/or post exercise recovery.

### 4.1. Baseline vs. Post-Intervention Measures

In the current study, participants completed the PSQI and RESTQ at baseline and post-intervention to assess changes in sleep quality and recovery/stress balance. At baseline, 87% (*n* = 13) of athletes were classified as poor sleepers (global PSQI score ≥ 5), which was consistent with previous research in elite athletes [3,29,30,31]. A growing body of research has highlighted the prevalence of sleep problems in athletes including insomnia symptoms [10,32] and obstructive sleep apnea (OSA) [30]. In the current study, there was a significant reduction in mean global PSQI scores (6.47 ± 2.17 to 4.13 ± 1.19) from baseline to post-intervention. Global PSQI scores improved significantly at post-intervention with fewer athletes (33%; *n* = 5) being classified as poor sleepers. This change was clinically significant because there were less sleep problems observed post-intervention.

Specifically, PSQI sleep quality improved significantly from baseline to post-intervention (1.53 ± 0.84 to 0.27 ± 0.46). Previous research using both subjective [33] and objective measures [34] has suggested that elite athletes have inferior sleep quality compared to non-athletes. Poor sleep quality is of particular concern for elite athletes as it can result in a reduction in recovery and/or subsequent athletic performance [4,35,36,37,38]. While there were improvements from baseline to post-intervention in PSQI dimensions of sleep onset latency (1.67 ± 0.49 to 1.35 ± 0.62), sleep duration (0.34 ± 0.49 to 0.14 ± 0.35) and sleep efficiency (0.6 ± 0.91 to 0.2 ± 0.42), no significant differences were observed. Similarly, daytime dysfunction component scores did not change from baseline to post-intervention, indicating no impact on levels of daytime sleepiness from baseline to post-intervention.

No participants reported using sleep medication at baseline or post-intervention. However, the small sample size of the current study (*n* = 15) must be noted. This is in stark contrast to a recent investigation in Finland (*n* = 228) which demonstrated that 33.9% (*n* = 76) used sleep medication [38]. A report from the National Collegiate Athletic Association (NCAA) indicated that sleep medication use accounted for 10.3% of miscellaneous substance use across all sports in American student athletes [39]. The lack of sleep medication usage in the current study, despite 87% (*n* = 13) of the athletes reporting poor sleep at baseline, highlights the potential need for evidence-based nutritional interventions and protocols (e.g., kiwifruit) to promote sleep health in elite athletes.

In the current study, there were statistically significant improvements from baseline to post-intervention for the RESTQ scale general stress (3 ± 0.86 vs. 2.58 ± 0.58) and sport stress (2.72 ± 0.65 vs. 2.39 ± 0.63). Compared to previous research in Rugby players (*n* = 41), general stress (forwards 1.38 ± 0.62 and backs 1.57 ± 0.68) and sport stress (forwards 1.26 ± 0.51 and backs 1.67 ± 0.73) scores in the current study were higher (i.e., indicating more stress) at baseline and post-intervention. However, it must be noted that this sample were student athletes and not necessarily competing at the same level as the participants in the current study [40].

In terms of the 19 RESTQ sub scale items from baseline to post-intervention, there were statistically significant reductions in fatigue (3.4 ± 1.42 vs. 2.8 ± 1.21), physical complaints (2.37 ± 0.92 vs. 1.8 ± 0.6) and disturbed breaks (2.38 ± 0.8 vs. 2.03 ± 0.75). Significant associations between the RESTQ subscales fatigue (OR 1.7) and disturbed breaks (OR 1.84) have been demonstrated in German professional soccer players (*n* = 22) suggesting that injury risk increased due to insufficient rest periods and/or if players felt exhausted or overtrained [41]. The RESTQ scores at baseline suggested that the athletes in the current study would benefit from an intervention aimed at promoting sleep and/or recovery, while the changes in the RESTQ scores from baseline to post-intervention suggest a small but potentially meaningful change in athletes’ recovery stress balance.

#### 4.1.1. Over the Course of the Intervention—Sleep Diary

To assess the impact of kiwifruit consumption on sleep quality, sleep duration, fatigue and sleep health, for the duration of the intervention (1 baseline week and 4 intervention weeks), participants completed the CSD-C including additional questions relating to fatigue. It has recently been suggested that elite athletes are prone to sleep inadequacies characterised by habitual short sleep duration (<7 h/night), unrefreshing sleep, long SOL, daytime sleepiness, daytime fatigue and poor sleep quality [1]. In the current study, mean TST (hours) improved from baseline (7.6 ± 0.75) to week 2 (8.4 ± 0.62), week 3 (8.42 ± 0.34), week 4 (8.55 ± 0.44) and week 5 (8.63 ± 0.47); mean values week to week met current sleep guidelines (i.e., 7–9 h) for adults. Overall, the mean TST moved from inadequate during the baseline week to within the recommended 8–10 h range for athletes during the intervention. However, it must be noted that the athletes self-reported their sleep behaviours using a sleep diary, which can be affected by recall bias, e.g., overestimation of sleep duration and efficiency [1]. Shorter sleep durations can directly impact athletic performance through negative effects on the heart rate, breathing rate and lactate concentrations [42] or indirectly through an impact on mood, motivation or rate of perceived exertion (RPE) [43,44]. The amount of sleep an individual habitually obtains has implications for their ability to function effectively [45]. Hence, improvements in sleep duration as seen in the current study could positively impact health and performance.

A recent study which assessed sleep using actigraphy in a large elite athlete sample (*n* = 175) demonstrated that habitual sleep duration was 6.7 ± 0.8 h while self-identified sleep need was 8.3 ± 0.9 h and suggested that individual athletes sleep less than team sport athletes [45]. It is possible that some athletes require <7–9 h sleep while others require more [45]. More research is necessary to gain an understanding of the sleep needs of athletes, how often athletes achieve these sleep needs and possible interventions (e.g., nutrition) than can positively impact athlete sleep.

Previous research has suggested that there are no differences between the sexes for habitual sleep duration [34,45]. However, it must be noted that female athletes tend to be under-represented in research investigating the sleep and recovery of athletes, and these comparisons are not sport-specific. Differences in male and female physiology and biochemistry have been established, e.g., males typically have greater muscle mass and less adipose tissue which contributes to greater strength, aerobic and anaerobic power compared to females [46,47]. The impact of alterations in female sex hormone concentrations during the menstrual cycle on sleep and recovery of elite athletes warrants further investigation. In the current study, >50% (*n* = 8) of the sample were female athletes, and there were no significant gender or sport effects on measures of sleep duration or quality. However, further research that focuses on gender differences within sports is warranted as research in athletes tends to focus on comparisons among athlete groups rather than comparison within specific sports.

It has recently been suggested sleep fragmentation is a contributing factor to poor sleep quality in athletes [1]. It is estimated that athletes need 8.3 ± 0.9 h sleep [45]; the increase in TIB during the intervention would increase the likelihood of an athlete achieving their sleep need. Awakenings reduced from baseline (1.22 ± 0.87) to intervention with significant reduction in week 3 (0.89 ± 0.94) and week 4 (0.95 ± 0.96). WASO also reduced from baseline and significantly in week 3 (10.8 ± 10.2), week 4 (5.56 ± 4.43) and week 5 (5.68 ± 5.19).

Good sleep quality is recognised as a predictor of physical health, mental health and wellness, while poor sleep quality can lead to fatigue, drowsiness and changes in mood [46]. Although SE is a good starting point in terms of sleep improvement, athletes also need to focus on sleep quality [47]. Sleep quality can be difficult to assess, especially in athletes [47]. However, it has been recommended that sleep efficiency should be used to monitor sleep quality using actigraphy in athletes [48]. Previous research has highlighted that athletes’ sleep quality as measured by SE (%) was lower (3–4%) during the night before competition [43]. Differences have also been observed in the sleep characteristics of team sport and individual athletes whereby individual athletes had poorer sleep efficiency than team sport athletes [49]. In the current study, SE (%) increased significantly from baseline (86.2 ± 5.31) to week 2 (91.5 ± 3.8), week 3 (93.4 ± 2.7), week 4 (93 ± 2.54) and week 5 (93.3 ± 2.43) which is reflective of the increased TST and/or reductions in WASO and SOL. SE < 85% is considered poor [50]; in the current study, baseline SE (%) scores straddled and, for a minority of athletes (*n* = 4), were below the threshold of 85%, indicating insomnia symptoms [51]. The improvement in SE (%) observed from throughout the intervention resulted in less insomnia symptomology among the sample but could also impact performance. Insufficient sleep has been negatively associated to physical performance (speed and anaerobic power), neurocognitive function (attention and memory) and physical health (illness and injury risk) [46,51,52,53]. The scores from the current study are similar to previous research which has demonstrated that the habitual sleep efficiency of elite athletes was 88.47% ± 5.45% [29], 80.6% ± 6.4% [34], 86.3% ± 6.1% [49] and 79% ± 9.2% [54]. A recent systematic review reported the pooled average sleep efficiency for athletes (86% ± 5%; range 79–96%) [10].

Self-reported Fatigue Going to Bed did not differ significantly from baseline to intervention. However, there was a significant reduction in Fatigue in the Morning from baseline to week 5, which coincided with the improvements reported in the sleep diaries. The improvement in Fatigue in the Morning is beneficial as sleep problems in athletes have been noted previously. A recent systematic review demonstrated the prevalence of insomnia symptomology (i.e., increased SOL, greater sleep fragmentation, non-restorative sleep and excessive daytime fatigue) [10]. While no significant difference was observed, Sleep Quality scores improved during the intervention compared to baseline. These improvements in sleep duration and quality highlight kiwifruit as a potential athlete-friendly intervention that could promote improve sleep and recovery.

#### 4.1.2. Over the Course of the Intervention—Sleep Health

The RU-SATED has demonstrated adequate internal consistency (Cronbach’s α = 0.64) [27], most likely due to the low number of items (6), as the size of alpha depends on the number of items in a scale [55]. However, mean inter-item correlations (r = 0.29–0.5) were moderate (27), and it has previously been suggested that inter-item correlations should fall between 0.15–0.5 [56]. The RU-SATED is a valid instrument for the assessment of sleep health in adults that is related to but distinct from other sleep constructs [27].

To assess the impact of kiwifruit consumption on sleep quality, sleep duration, fatigue and sleep health for the duration of the intervention (1 baseline week and 4 intervention weeks), participants completed the RU-SATED. Poor sleep health can impair physical health; recently, it was demonstrated that students with poor sleep health were more likely to have poor physical health [57]. Sleep health in athletes is characterised by good sleep quality, minimal daytime dysfunction, strategic napping if necessary and good sleep hygiene [6]. In the current study, the participants reported relatively good sleep health scores at baseline. As a result, there were no significant differences from baseline to intervention for Regulation, Satisfaction/Quality, Timing and Duration. SE increased significantly from baseline (2.37 ± 0.66) to intervention, and there was a significant improvement in week 3 (2.47 ± 0.65), week 4 (2.46 ± 0.65) and week 5 (2.46 ± 0.62) similar to the PSQI sleep efficiency scores. Acute sleep deprivation and sleep disturbance (short sleep duration or reduced sleep efficiency) can impact immunity, which has been attributed to reduced growth hormone release during deep sleep and increased sympathetic output [58]. Reduced growth hormone release could also negatively impact athlete recovery following training or competition.

The results of this study suggest that consuming two kiwifruit one hour before bed is a wholefood-based intervention that has the potential to promote sleep and recovery in athletes. Further research is warranted in athletic populations to investigate the impact of kiwifruit consumption on sleep and recovery.

### 4.2. Limitations

This study is a novel investigation of the impact of a wholefood nutrition intervention (2 × kiwifruit 1 h before bed) on the sleep and recovery of elite athletes. Two elite athlete squads were recruited for this research. However, the sample size was small (*n* = 15; *n* = 9 sailing and *n* = 6 athletics), but it must be noted that the sample represented all the members of both squads. It is recognised that the recruitment and retention of elite athletes for research can be difficult, but it is essential as this research informs evidence-based practice [59]. The RU-SATED sleep health scores increased from baseline to weeks 2, 3, 4 and 5, indicating an improvement in sleep health during the intervention. However, the differences were not significant, possibly due to the small sample size or a ceiling effect.

Another limitation of the current study is the reliance on self-report measures for sleep and recovery. As noted previously, self-report measures (i.e., questionnaires and diaries) are prone to measurement error and recall bias [28], and athletes may overestimate their sleep duration [60,61]. However, self-report measures are accepted within athletic settings, as they are a relatively simple and inexpensive approach to athlete monitoring, affording a more representative overview of the target population [62], and subjective sleep tends to relate to complaints and help-seeking behaviour. Future research into the potential role of kiwifruit supplementation in the facilitation of athlete sleep and recovery should incorporate both subjective and objective measures of sleep.

The absence of objective measures of sleep (e.g., PSG, actigraphy) must be acknowledged as a limitation, but unfortunately, such measures were not feasible as the research was conducted during lockdown as a result of the global COVID-19 pandemic. The entire study was conducted during lockdown, and the pandemic severely curtailed research as countries were forced to go into lockdown as the virus spread [63]. Similar to the current study, the majority of research during the pandemic had to be modified to facilitate data collection and maintain participant risk of COVID-19 infection [63]. When it is possible to do so, this study should be replicated using a larger cohort of elite athletes incorporating a combination of subjective and objectives measures of sleep and recovery in a randomised control trial.

### 4.3. Practical Applications

The potential roles for specific foods and/or nutrients in promoting sleep quantity or quality and athlete recovery are an emerging area of interest within sport nutrition research. Potential nutritional interventions that could positively impact athletes’ sleep and cause resultant improvements in recovery warrant investigation. The potential of nutrition to influence sleep is related to various neurotransmitters associated with the sleep–wake cycle (e.g., melatonin and serotonin in kiwifruit). Although the research in this field is in its infancy, the current study adds to the limited body of evidence that kiwifruit consumption can positively impact sleep. The manipulation of the timing and dose of kiwifruit may have applications in terms of sleep and recovery (e.g., antioxidant consumption in relation to training) in athletes that warrant further investigation. The results presented suggest a potential role for kiwifruit consumption in sleep promotion and recovery protocols for elite athletes. Consuming 2 kiwifruit 1 h before bed is a practical wholefood-based intervention that can easily be implemented in real-world settings. Kiwifruit is available in wholefood form, but it is also consumed in various processed forms, e.g., drinks, sweets, lyophilised products (i.e., freeze dried), dehydrated products and juices [64]. Further research is warranted to develop protocols and/or products designed specifically to promote sleep and/or recovery in elite athletes.

## 5. Conclusions

The consumption of two kiwifruit one hour before bed for four weeks has the potential to positively impact the sleep and recovery of athletes. The results of the current study demonstrated a positive impact of kiwifruit consumption on key aspects of sleep and recovery in elite athletes. In summary, from baseline to post-intervention, there were clinically significant improvements in sleep quality (i.e., improved PSQI global scores and sleep quality component scores) and improvements in recovery stress balance (i.e., reduced general stress and sports stress scales). During the intervention, consumption of two kiwifruit one hour before bed improved sleep as evidenced by significant increases in TST and SE % and significant reductions in the number of awakenings and WASO.

## Figures and Tables

**Figure 1 nutrients-15-02274-f001:**
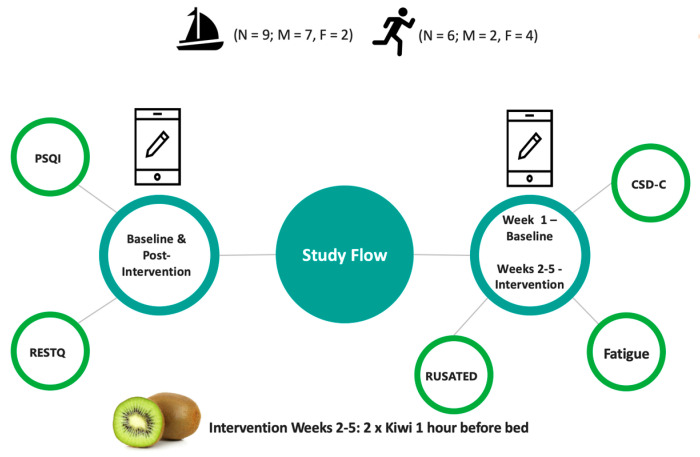
Study flowchart.

**Figure 2 nutrients-15-02274-f002:**
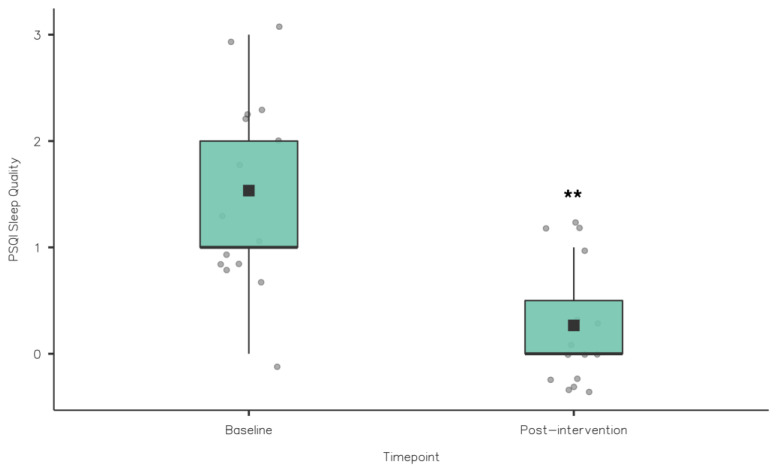
PSQI Sleep Quality Baseline vs. Post-Intervention (** *p* ≤ 0.01).

**Figure 3 nutrients-15-02274-f003:**
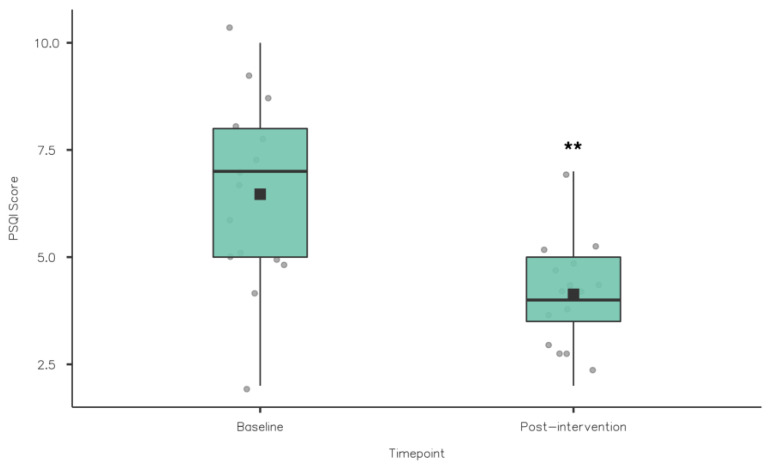
PSQI Global scores baseline vs. post-intervention (** *p* ≤ 0.01).

**Figure 4 nutrients-15-02274-f004:**
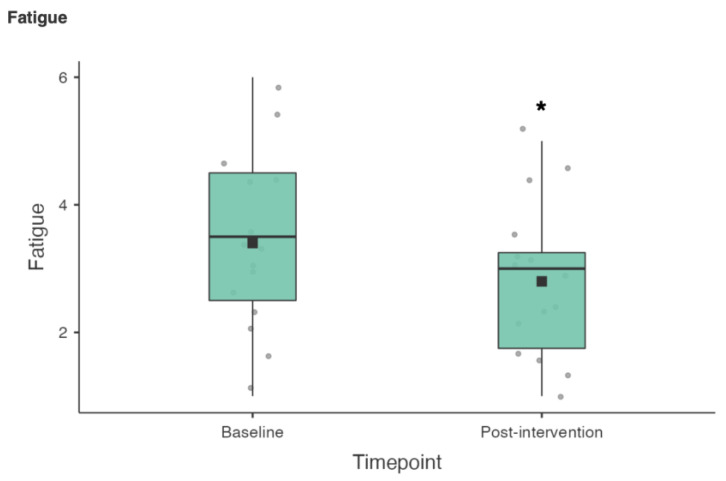
RESTQ Fatigue scores baseline vs. post-intervention (* *p* < 0.05).

**Figure 5 nutrients-15-02274-f005:**
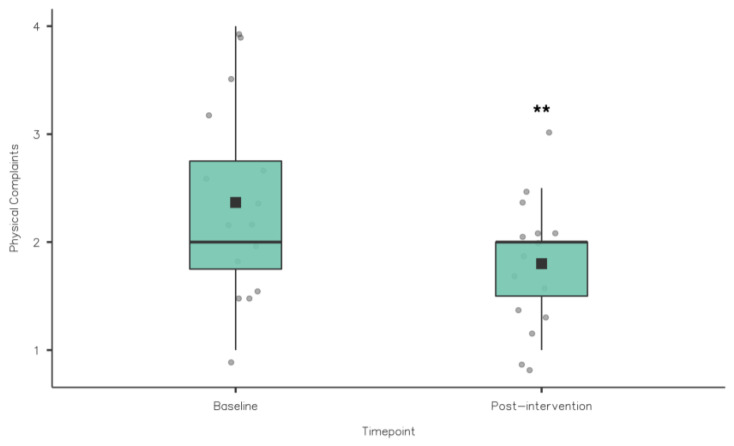
RESTQ Physical Complaints scores baseline vs. post-intervention (** *p* ≤ 0.01).

**Figure 6 nutrients-15-02274-f006:**
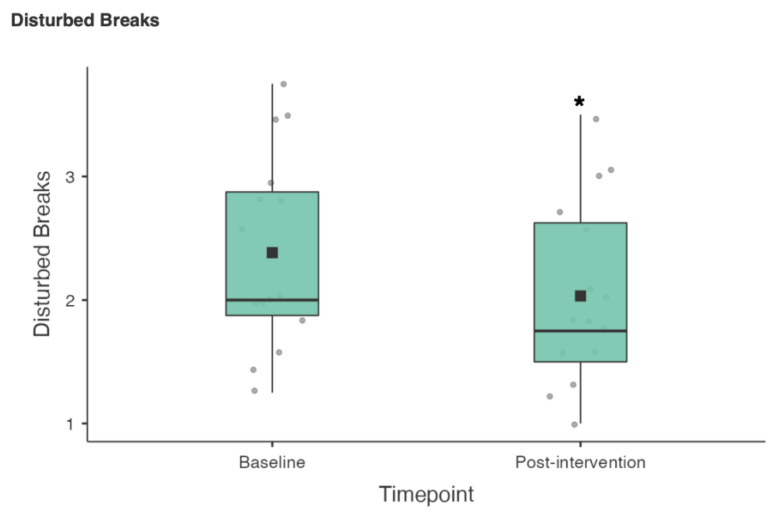
RESTQ Disturbed Breaks scores baseline vs. post-intervention (* *p* < 0.05).

**Figure 7 nutrients-15-02274-f007:**
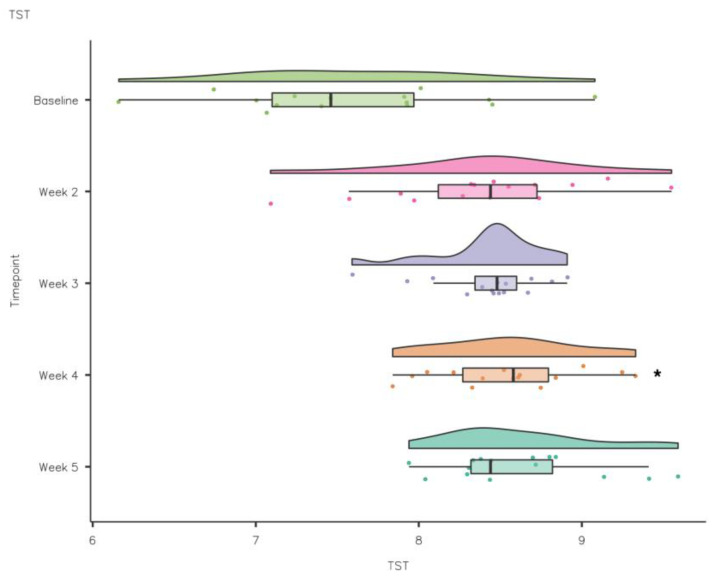
CSD-C Total Sleep Time data comparison week by week (* *p* < 0.05).

**Figure 8 nutrients-15-02274-f008:**
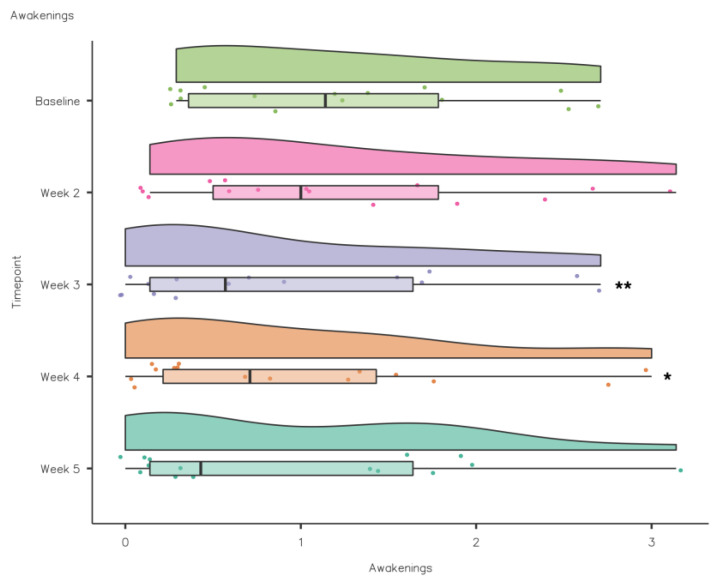
CSD-C Awakenings data comparison week by week (** *p* ≤ 0.01; * *p* < 0.05).

**Figure 9 nutrients-15-02274-f009:**
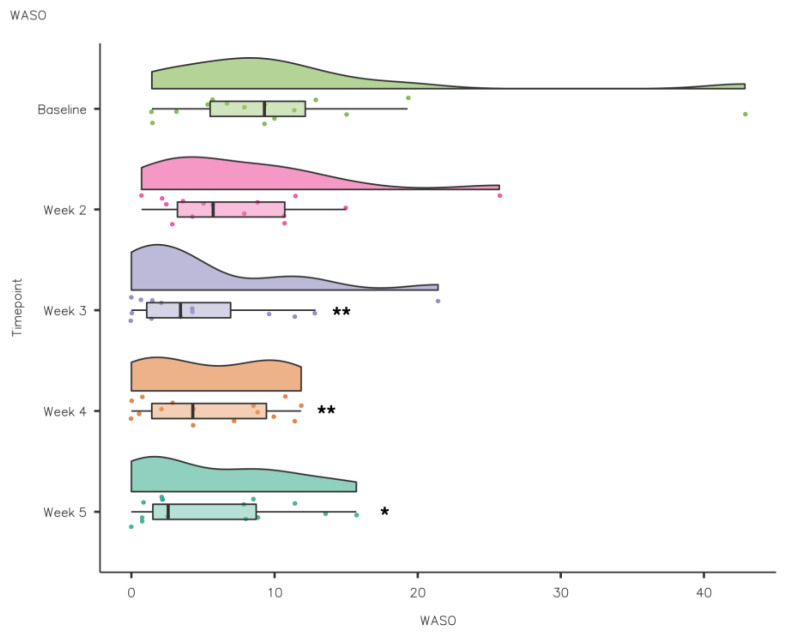
CSD-C WASO data comparison week by week (** *p* ≤ 0.01; * *p* < 0.05).

**Figure 10 nutrients-15-02274-f010:**
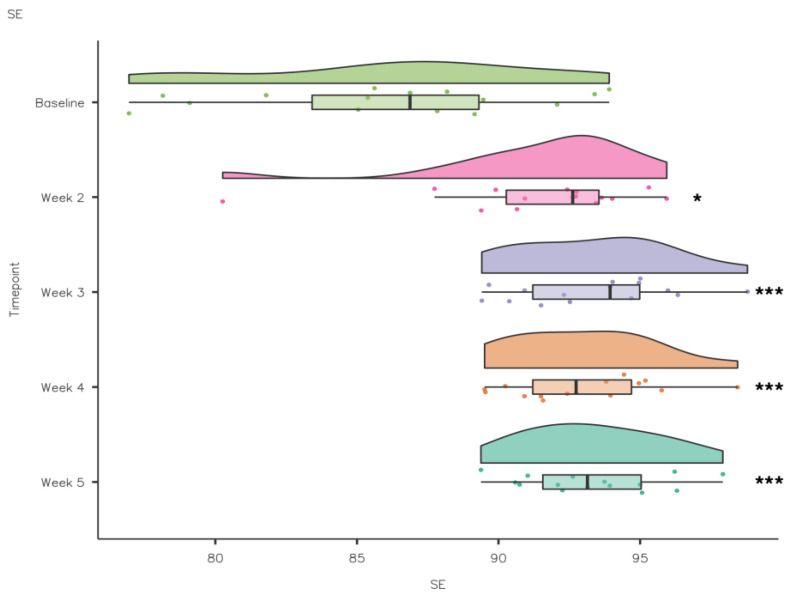
CSD-C Sleep Efficiency data comparison week by week (* *p* < 0.05; *** *p* ≤ 0.001).

**Figure 11 nutrients-15-02274-f011:**
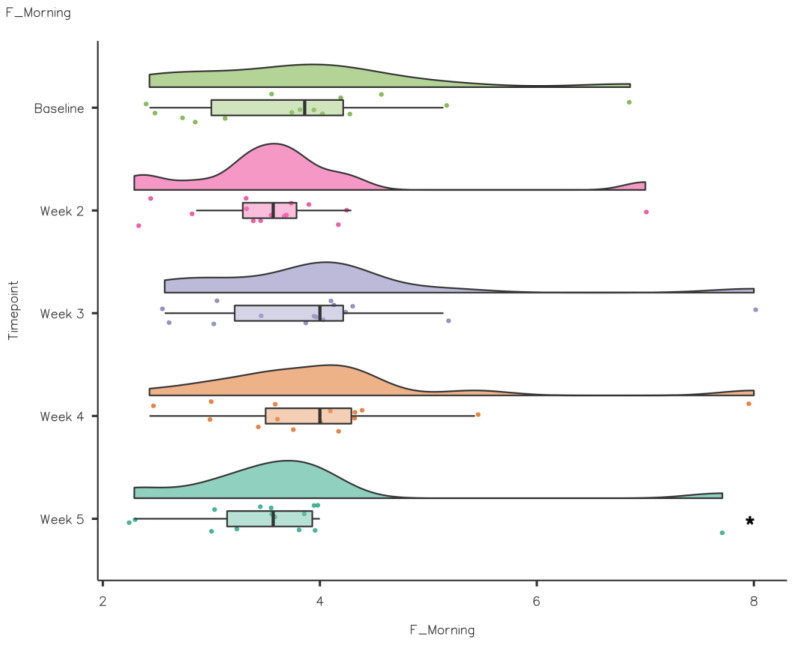
CSD-C Fatigue in the Morning data comparison week by week (* *p* < 0.05).

**Figure 12 nutrients-15-02274-f012:**
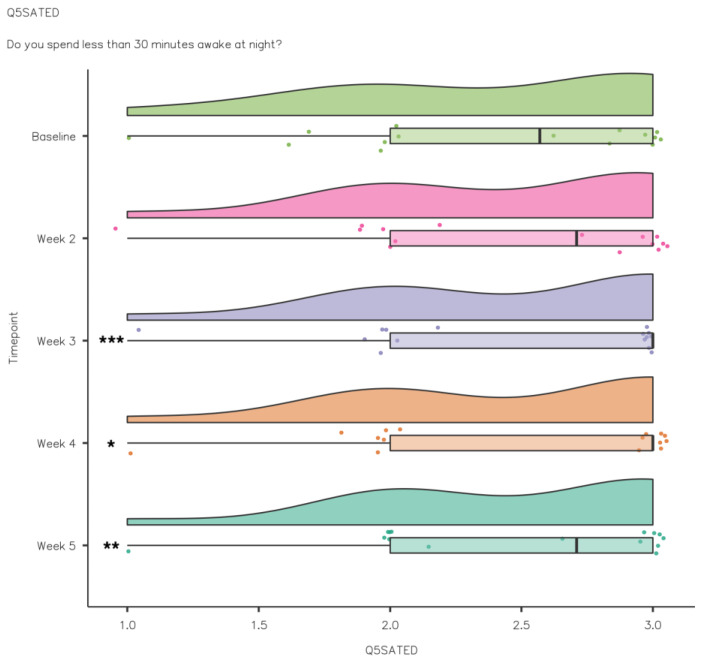
RU-SATED Efficiency data comparison week by week (*** *p* ≤ 0.001; ** *p* ≤ 0.01; * *p* < 0.05).

**Table 1 nutrients-15-02274-t001:** Characteristics of participants.

	All	Sailing	Athletics
**Gender**	15 (M = 9/F = 6)	9 (M = 7/F = 2)	6 (M = 2/F = 4)
**Age (Y)**	23.2 ± 3.9	24.56 ± 4	21.17 ± 2.93
**Body mass (kg)**	70.39 ± 13.34	78.89 ± 7.42 ***	57.65 ± 9.25
**Height (cm)**	175.37 ± 8.99	179.22 ± 7.31 *	169.58 ± 8.58
**Phase of season**	Pre-season *n* = 10 Competition *n* = 2 Off-season *n* = 3	Pre-season *n* = 6 Competition *n* = 1 Off-season *n* = 2	Pre-season *n* = 4 Competition *n* = 1 Off-season *n* = 1
**Normal training time**	8 a.m. to 5 p.m. *n* = 14 After 5 p.m. *n* = 1	8 a.m. to 5 p.m. *n* = 9	8 a.m. to 5 p.m. *n* = 5 After 5 p.m. *n* = 1
**Training/competition duration per week (mins)**	912 ± 359.19	636.67 ± 237.8 **	1095.56 ± 309.32

Statistically significant *** *p* ≤ 0.001; ** *p* ≤ 0.01; * *p* < 0.05.

**Table 2 nutrients-15-02274-t002:** Comparison of global PSQI score and component scores (mean ± SD) baseline vs. post-intervention.

	Baseline	Post-Intervention	Mean Difference	95% CI	Effect Size	*p*-Value
**Sleep Quality**	1.53 ± 0.84	0.27 ± 0.46	1.27	1–2	1	0.002 **
**Sleep Latency**	1.67 ± 0.49	1.33 ± 0.62	1	−1.79–1	0.46	0.18
**Sleep duration**	0.34 ± 0.49	0.14 ± 0.35	1	−0.02–0.43	1	0.15
**Sleep Efficiency**	0.6 ± 0.91	0.2 ± 0.42	1.5	1–2	1	0.09
**Sleep Disturbance**	1.2 ± 0.56	1.2 ± 0.56	0	0	0	1
**Medication**	0	0	0	0	0	0
**Daytime Dysfunction**	1.13 ± 0.74	1 ± 0.54	4.31	−1.98–1	0.33	0.48
**PSQI Global Score**	6.47 ± 2.17	4.13 ± 1.19	2.5	1.5–3.5	1	0.002 **

Statistically significant ** *p* ≤ 0.01.

**Table 3 nutrients-15-02274-t003:** Recovery Stress (Mean ± SD) Subscales Baseline vs. Post-Intervention.

	Baseline	Post-Intervention	Mean Difference	95% CI	Effect Size	*p*-Value
**General Stress**	2.83 ± 1.54	2.37 ± 1.01	0.5	−0.13–1.06	0.48	0.16
**Emotional Stress**	2.9 ± 0.95	2.47 ± 0.61	0.5	−2.71–1.25	0.44	0.17
**Social Stress**	3.03 ± 1.03	2.57 ± 0.84	0.75	−0.5–1.75	0.64	0.95
**Conflicts/Pressure**	3.43 ± 1.31	3.17 ± 0.92	0.75	−0.5–1.75	0.5	0.3
Fatigue	3.4 ± 1.42	2.8 ± 1.21	1	0.5–1.5	0.84	0.02 *
**Lack of Energy**	3.07 ± 1.18	2.87 ± 1.09	0.5	−4.26–1	0.56	0.15
**Physical Complaints**	2.37 ± 0.92	1.8 ± 0.6	0.75	0.5–1.25	1	0.005 **
**Success**	3.47 ± 0.9	3.73 ± 0.93	−0.5	−1.5–0.5	−0.43	0.4
**Social Recovery**	4.13 ± 1.1	4.43 ± 1.1	−0.68	−1.25–0.5	−0.61	0.14
**Physical Recovery**	3.53 ± 1.19	3.83 ± 1.18	−0.5	−1–0.25	−0.5	0.17
**General Wellbeing**	4.47 ± 1.27	4.5 ± 0.95	−2.7	−1.5–1.25	−0.02	1
**Sleep Quality**	3.57 ± 1.22	4.07 ± 1.35	−1	−1.75–0.5	−0.64	0.1
**Disturbed Breaks**	2.38 ± 0.8	2.03 ± 0.75	0.5	5.9–0.88	0.73	0.04 *
**Emotional Exhaustion**	2.7 ± 1.14	2.33 ± 0.98	0.5	−2.3–1	0.67	0.051
**Injury**	3.08 ± 0.68	2.8 ± 0.94	0.38	−0.13–0.75	0.42	0.2
**Being in Shape**	4.23 ± 1.25	4.4 ± 1.1	−0.25	−1–0.5	−0.37	0.33
**Personal Accomplishment**	3.58 ± 1.08	3.7 ± 1.13	−0.16	−0.5–0.38	−0.35	0.3
**Self-Efficacy**	3.95 ± 1.27	3.93 ± 1.09	6.01	−0.75–0.75	0.03	0.96
**Self-Regulation**	4.33 ± 1.15	4.32 ± 0.84	0.13	−0.75–0.88	0.13	0.77

Statistically significant ** *p* ≤ 0.01; * *p* < 0.05.

**Table 4 nutrients-15-02274-t004:** Sleep diary comparison week by week (mean ± SD).

	Baseline	Week 2	Week 3	Week 4	Week 5
**SOL (mins)**	24.6 ± 15.9	18.8 ± 11	14.9 ± 9.51	16.1 ± 10.3	13.9 ± 8.4
**Awakenings**	1.22 ± 0.87	1.2 ± 0.98	0.89 ± 0.94 **	0.95 ± 0.96 *	0.98 ± 0.96
**WASO (mins)**	10.8 ± 10.2	7.8 ± 6.45	5.15 ± 6.12 **	5.56 ± 4.43 **	5.68 ± 5.19 *
**TIB (h)**	8.84 ± 0.75	9.18 ± 0.5	9.02 ± 0.41	9.2 ± 0.54	9.24 ± 0.47
**TST (h)**	7.6 ± 0.75	8.4 ± 0.62	8.42 ± 0.34	8.55 ± 0.44 *	8.63 ± 0.47
**SE (%)**	86.2 ± 5.31	91.5 ± 3.8 *	93.4 ± 2.7 ***	93 ± 2.54 ***	93.3 ± 2.43 ***
**Fatigue (Bed)**	3.23 ± 0.78	3.1 ± 0.58	3.28 ± 0.57	3.24 ± 0.49	3.16 ± 0.69
**Fatigue (Morning)**	3.85 ± 1.15	3.67 ± 1.08	4.03 ± 1.31	4.1 ± 1.3	3.7 ± 1.24 *
**Sleep Quality**	3.45 ± 0.74	3.48 ± 0.78	3.76 ± 0.7	3.76 ± 0.68	3.58 ± 0.83

Statistically significant difference (* *p* < 0.05; ** *p* < 0.01; *** *p* ≤ 0.001).

**Table 5 nutrients-15-02274-t005:** Sleep health comparison week by week (mean ± SD).

	Baseline	Week 2	Week 3	Week 4	Week 5
**Regulation**	2.6 ± 0.46	2.56 ± 0.47	2.58 ± 0.48	2.62 ± 0.46	2.59 ± 0.52
**Satisfaction/Quality**	2.33 ± 0.51	2.27 ± 0.61	2.25 ± 0.68	2.3 ± 0.62	2.31 ± 0.62
**Alertness/Sleepiness**	2.78 ± 0.36	2.83 ± 0.31	2.88 ± 0.28	2.9 ± 0.26	2.9 ± 0.27
**Timing**	2.98 ± 0.08	2.97 ± 0.08	2.99 ± 0.04	2.98 ± 0.07	2.99 ± 0.04
**Efficiency**	2.37 ± 0.66	2.43 ± 0.63	2.47 ± 0.65 ***	2.46 ± 0.65 *	2.46 ± 0.62 **
**Duration**	2.81 ± 0.36	2.75 ± 0.43	2.74 ± 0.43	2.74 ± 0.45	2.74 ± 0.44
**Sleep Health Score**	9.88 ± 1.63	9.81 ± 1.75	9.9 ± 1.65	9.97 ± 1.68	9.98 ± 1.7

Statistically significant *** *p* ≤ 0.001; ** *p* ≤ 0.01; * *p* < 0.05.

## Data Availability

The data presented in this study are available on request from the corresponding author.

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
