# Peer review of "The Impact of Kiwifruit Consumption on the Sleep and Recovery of Elite Athletes"

_nutrients, 2023, doi:10.3390/nu15102274_

Round 1

Reviewer 1 Report

The manuscript is prepared relatively correctly.  However, there are a few critical issues that should be addressed before possible publication acceptation.

In this manuscript, the authors attempted to examine the effect of Kiwifruit consumption on sleep quality and quantity in high-performance athletes. This is a topic that interests scientists, coaches and athletes and deserves further research. Due to the increasing attention to proper nutrition in athletes, this problem is very important. However, in its present form, many shortcomings prevent publication of this manuscript in a Journal. My specific comments are as follows:

Introduction. Please indicate the average content of melatonin and serotonin in Kiwifruit. Are there studies determining the differences in the content of the described substances between individual varieties of Kiwifruit?

What percentage of dopamine, serotonin contained in Kiwi fruit are absorbed in the intestines. Do exogenous substances cross the blood-brain barrier?

Section 2.2. Please list what sports the athletes practiced (sprints, middle- and long-distance?, technical - for example: javelin?).

The authors in section 2.2. reported that the athletes did not consume sleep medication. Did they take any supplements (BCAA, ZMA, GABA) that could potentially affect the amount and quality of sleep?

Table. 1. Phase of season - please check the size of individual groups. I think a mistake has been made. Values for Sailing and Athletics are swapped.

Table. 1. Normal training time: replace 8am -5pm with: 8am to 5pm

Table. 1. There is no statistically significant difference of the age between groups?

3. Results - what does it mean see Table 6.1???? – line 203

Line 217 - what does figure 6.3 mean

Why do the authors repeat the values of the obtained results three times, i.e. in the text (line 216-217), table 2 and figure 3. For clarity, please select one option. This is where the table seems most reasonable. In general, this remark applies to the entire manuscript, so authors should choose to present the most representative way of presenting the results.

Identical as above: line 215, table 2 and figure 2

Line 245 - what does Figure 4-6 mean?

Line 260 - what does Table 6-4 mean?

Line 275 – differemnce?

Figure 9 - add unit to TST (hours)

In general, add units in figures and expand abbreviations used in these figures. In my opinion, you should point out that the reported statistically significant differences (* or ** or ***) refer to the measurement relative to Baseline under the tables and figures.

The authors do not mention whether the surveys included possible naps (daytime sleep)?

The authors declare that in the questionnaires they specified dairy intake. Why didn't they show these results at work? On the other hand, what is the additional effect of this dietary component on sleep?

Why nutritional questionnaire were not conducted?

Shouldn't the count of Kiwifruit be adjusted to the weight of athletes?

How the trainings were realized during the lockdown period?

Didn't the mere fact of the pandemic and the resulting stress affect the amount and quality of sleep?

In the discussion, the authors should present other research results with the effect of other fruit consumption on sleep quality, not necessarily with Kiwifruit.

Author Response

Thank you for the thorough and comprehensive review, please see the responses attached.

The impact of Kiwifruit consumption on the sleep of elite athletes.

Response to Reviewer 1

Author responses in green.

The manuscript is prepared relatively correctly.  However, there are a few critical issues that should be addressed before possible publication acceptation.

Thank you for a very thorough and constructive review of our manuscript.

Introduction. Please indicate the average content of melatonin and serotonin in Kiwifruit. Are there studies determining the differences in the content of the described substances between individual varieties of Kiwifruit?

Lines 57-59 have been edited to reflect this comment: “Kiwifruit have also been shown to contain melatonin (24 µg/g)  [16], which plays an important role in circadian rhythm regulation i.e. getting to sleep and maintaining sleep are easiest at and after the onset of melatonin secretion. The Serotonin (5.8 µg/g) content…”

The difference in content between varieties of Kiwifruit was not discussed as one variety was consumed i.e. Green Kiwifruit (Actinidia Deliciosa).

What percentage of dopamine, serotonin contained in Kiwi fruit are absorbed in the intestines. Do exogenous substances cross the blood-brain barrier?

Thank you for this suggestion but this information is outside the scope of the current study.

Section 2.2. Please list what sports the athletes practiced (sprints, middle- and long-distance?, technical - for example: javelin?).

Lines 109-111 have been edited to address this comment: A group of elite athletes (n=15) from a national sailing squad (n = 9; 7 males and 2 females) and a national athletics squad (middle distance runners; n = 6; 2 males and 4 females; see table 1) were recruited through the National Sports Institute.

The authors in section 2.2. reported that the athletes did not consume sleep medication. Did they take any supplements (BCAA, ZMA, GABA) that could potentially affect the amount and quality of sleep?

No participants reported using BCAAs, ZMA and/or GABA at baseline or post-intervention.

Table. 1. Phase of season - please check the size of individual groups. I think a mistake has been made. Values for Sailing and Athletics are swapped.

Thank you, we have made this change to Table 1.

Table. 1. Normal training time: replace 8am -5pm with: 8am to 5pm

Thank you, we have made this change to Table 1.

Table. 1. There is no statistically significant difference of the age between groups?

No, the difference was not statistically significant.

  1. Results - what does it mean see Table 6.1???? – line 203

The sailors were heavier and taller which is indicative of their sport compared to middle distance running.

Line 217 - what does figure 6.3 mean

This is explained in section 3.1.1 Sleep Quality (Lines 222-224): PSQI global scores reduced significantly from baseline (6.47 ± 2.17) to post-intervention (4.13 ± 1.19; z=91, p = 0.002) (see Table 2 and Figure 6.3), this was also clinically relevant indicating a significant reduction in sleep problems among the athletes.

Why do the authors repeat the values of the obtained results three times, i.e. in the text (line 216-217), table 2 and figure 3. For clarity, please select one option. This is where the table seems most reasonable. In general, this remark applies to the entire manuscript, so authors should choose to present the most representative way of presenting the results.

This is one of the most significant findings within the study and warrants representation in both the table and a figure.

Identical as above: line 215, table 2 and figure 2

This is one of the most significant findings within the study and warrants representation in both the table and a figure.

Line 245 - what does Figure 4-6 mean?

This is outlined in the discussion (Lines 383-392): “In terms of the 19 RESTQ sub scale items from baseline to post-intervention, there were statistically significant reductions in fatigue (3.4 ± 1.42 vs. 2.8 ± 1.21), physical complaints (2.37 ± 0.92 vs. 1.8 ± 0.6) and disturbed breaks (2.38 ± 0.8 vs. 2.03 ± 0.75). Significant associations between the RESTQ subscales fatigue (OR 1.7) and disturbed breaks (OR 1.84) have been demonstrated in German professional soccer players (n=22) suggesting that injury risk increased due to insufficient rest periods and/or if players feel exhausted or overtrained [41]. The REST-Q scores at baseline suggested that the athletes in the current study would benefit from an intervention aimed at promoting sleep and/or recovery. While the changes in the REST-Q scores from baseline to post-intervention suggest a small but potentially meaningful change in athletes’ recovery stress balance.’

Line 260 - what does Table 6-4 mean?

This summarises the mean sleep diary figures by week.

Line 275 – differemnce?

Thank you this has been corrected.

Figure 9 - add unit to TST (hours)

All figures have been presented the same way units have not been included in the axis titles.

In general, add units in figures and expand abbreviations used in these figures. In my opinion, you should point out that the reported statistically significant differences (* or ** or ***) refer to the measurement relative to Baseline under the tables and figures.

We feel that the figures are presented in a clear and concise manner.

The authors do not mention whether the surveys included possible naps (daytime sleep)?

While it was an option in the sleep diary, no participants reported napping.

The authors declare that in the questionnaires they specified dairy intake. Why didn't they show these results at work? On the other hand, what is the additional effect of this dietary component on sleep?

There was a typological error on section 2.3 where ‘dairy’ had replaced ‘diary’, which has been corrected. Dairy intake was not specified in the current study.

Why nutritional questionnaire were not conducted?

To reduce participant burden as they had to complete a daily sleep diary which was time consuming.

Shouldn't the count of Kiwifruit be adjusted to the weight of athletes?

The dose was based on the doses employed in the two previous studies (2 x Kiwi [17]) and (130g [19]) in non-athletic populations

How the trainings were realized during the lockdown period?

Athletes were able to complete their normal training remotely and were in daily contact with their respective coaches.

Didn't the mere fact of the pandemic and the resulting stress affect the amount and quality of sleep?

Whilst the authors acknowledge the potential for ‘lockdowns’ to impact sleep, the inclusion of a baseline week was designed to mitigate this issue and all subjects reported maintenance of their ‘normal’ sleep patterns.

In the discussion, the authors should present other research results with the effect of other fruit consumption on sleep quality, not necessarily with Kiwifruit.

This was not done in order to maintain the focus of the discussion and these findings have been outlined extensively elsewhere. The use of kiwifruit is the novel aspect of this study and the authors wanted  to highlight the practical applications of this intervention.

Reviewer 2 Report

Thank you for allowing me to read this manuscript relating to a very interesting study. Overall, there is a solid rationale for the investigating and the findings are clearly presented. As such, I only have some minors comments that the authors may wish to consider…

Could the title reflect the key findings more, specifically that the ingestion of Kiwifruit positively effects sleep and recovery in elite athletes?

The abstract might benefit from the inclusion of a stronger rationale for the use of Kiwifruit, rather than general functional foods - Kiwifruit have been shown to contain melatonin, which plays an important role in circadian rhythm regulation. In addition, it would be helpful to support the results statements with some data.

The introduction provides a solid rationale for the study. Is there merit in including information for why two kiwifruits are required to be beneficial?

Lines 179-80: Due to participants purchasing the Kiwifruit, is there the potential for them to have received different sizes / doses depending on the size, ripeness, or age of the fruit?

It would be helpful to state what effect size was calculated, and the corresponding cut-offs in the methods section.

Is there any way of either joining the data points in the figures, or being able to identify the individuals so that the change for each participant can be seen? Otherwise, the data presented tells a very nice story.

Considering the coverage in the media, were the participants aware of the potential benefits of kiwifruit on sleep, and could this have affected the findings?

Considering that kiwifruit have been shown to have positive effects on digestion, could the potential alleviation of constipation, upper gastrointestinal (GI) symptoms such as abdominal discomfort and pain, indigestion, and reflux possibly contribute to the positive effects seen in the present study?

As acknowledged by the authors, data collection took place during lockdowns; is there potential for this to have affected the findings when significant disruptions were reported for all lifestyle factors including sleep patterns with increase in total sleep time and sleep latency, as well as a delay in mid-sleep times e.g., Facer-Childs et al. (2021)?

Are there any potential downsides to consuming kiwifruit before bed e.g. dental health? Or, as suggested by Graziani et al. (2018) kiwifruit consumption reduces gingival inflammation which might improve dental health, something that is often reported to be poor in athletes and has potential negative effects on performance.

I found this to be an interesting manuscript with, despite requiring further investigation, clear practical implications. I hope that the authors find the above comments to be helpful and in the constructive manner in which they are intended.

Author Response

Thank you for your fair and thorough review, please see the responses attached.

The impact of Kiwifruit consumption on the sleep of elite athletes.

Response to Reviewer 2

Authors’ responses in green.

Thank you for allowing me to read this manuscript relating to a very interesting study. Overall, there is a solid rationale for the investigating and the findings are clearly presented. As such, I only have some minors comments that the authors may wish to consider…

The authors would like to thank Reviewer 2 for a very fair and constructive review of our manuscript.

Could the title reflect the key findings more, specifically that the ingestion of Kiwifruit positively effects sleep and recovery in elite athletes?

We have edited the title of the paper to reflect the impact on recovery also: “The impact of Kiwifruit consumption on the sleep and recovery of elite athletes.”

The abstract might benefit from the inclusion of a stronger rationale for the use of Kiwifruit, rather than general functional foods - Kiwifruit have been shown to contain melatonin, which plays an important role in circadian rhythm regulation. In addition, it would be helpful to support the results statements with some data.

We have edited the abstract and included a rationale for Kiwifruit use and some key data.

The introduction provides a solid rationale for the study. Is there merit in including information for why two kiwifruits are required to be beneficial?

We have included are rationale for the kiwifruit dose in the Methodology (Lines 176-178): “The dose was based on doses employed in previous studies (2 x Kiwi [17]) and (130g [19]) and the timing was proposed to coincide with the melatonin secretion.”

Lines 179-80: Due to participants purchasing the Kiwifruit, is there the potential for them to have received different sizes / doses depending on the size, ripeness, or age of the fruit?

This was controlled as much as possible given the ;lockdown circumstances the study was conducted under. All athletes purchased the same variety of kiwifruit form the same supplier.

It would be helpful to state what effect size was calculated, and the corresponding cut-offs in the methods section.

We have added detail about how effect sizes were calculated and interpreted in the data analysis section (Lines 192-194): “Effect sizes were calculated using Cohen’s d, and interpreted as small d ≥ 0.2, medium d ≥ 0.5 and large d ≥ 0.8.”

Is there any way of either joining the data points in the figures, or being able to identify the individuals so that the change for each participant can be seen? Otherwise, the data presented tells a very nice story.

Unfortunately the analysis software used (Jamovi) does not allow for the data points to be joined.

Considering the coverage in the media, were the participants aware of the potential benefits of kiwifruit on sleep, and could this have affected the findings?

No participants were aware of the potential benefits of kiwifruit in relation to sleep and recovery.

Considering that kiwifruit have been shown to have positive effects on digestion, could the potential alleviation of constipation, upper gastrointestinal (GI) symptoms such as abdominal discomfort and pain, indigestion, and reflux possibly contribute to the positive effects seen in the present study?

As acknowledged by the authors, data collection took place during lockdowns; is there potential for this to have affected the findings when significant disruptions were reported for all lifestyle factors including sleep patterns with increase in total sleep time and sleep latency, as well as a delay in mid-sleep times e.g., Facer-Childs et al. (2021)?

Whilst the authors acknowledge the potential for ‘lockdowns’ to impact sleep, the inclusion of a baseline week was designed to mitigate this issue and all subjects reported maintenance of their ‘normal’ sleep patterns.

Are there any potential downsides to consuming kiwifruit before bed e.g. dental health? Or, as suggested by Graziani et al. (2018) kiwifruit consumption reduces gingival inflammation which might improve dental health, something that is often reported to be poor in athletes and has potential negative effects on performance.

All subjects were instructed to consume the kiwifruit 1 hour before bed and due the potential dental issues to brush their teeth before bed. The potential positive or negative impact off kiwifruit consumption on dental heath in elite athletes was beyond the scope of the current study but warrants further investigation.

I found this to be an interesting manuscript with, despite requiring further investigation, clear practical implications. I hope that the authors find the above comments to be helpful and in the constructive manner in which they are intended.

Thank you so much for a very thoughtful and fair review.